# AGL9: A Novel Hepatoprotective Peptide from the Larvae of Edible Insects Alleviates Obesity-Induced Hepatic Inflammation by Regulating AMPK/Nrf2 Signaling

**DOI:** 10.3390/foods10091973

**Published:** 2021-08-24

**Authors:** Meiqi Fan, Young-Jin Choi, Yujiao Tang, Ji Hye Kim, Byung-gyu Kim, Bokyung Lee, Sung Mun Bae, Eun-Kyung Kim

**Affiliations:** 1Division of Food Bioscience, College of Biomedical and Health Sciences, Konkuk University, Chungju 27478, Korea; fanmeiqi@kku.ac.kr; 2Department of Food Science and Nutrition, College of Health Science, Dong-A University, Busan 49315, Korea; choiyoung11@donga.ac.kr (Y.-J.C.); bolee@dau.ac.kr (B.L.); 3Center for Silver-Targeted Biomaterials, Brain Busan 21 Plus Program, Dong-A University, Busan 49315, Korea; 4School of Life Sciences, Changchun University of Science and Technology, Changchun 130600, China; yuanxi00@126.com; 5Gyeongnam Agricultural Research and Extension Services, Jinju 52733, Korea; xodnvos@gmail.com; 6Center for Genomic Integrity, Institute for Basic Science, Ulsan 44919, Korea; goldenlion@ibs.re.kr

**Keywords:** hepatoprotective peptide, AGL9, hepatic lipid metabolism, inflammation, nonalcoholic fatty liver disease

## Abstract

In this study, we investigated the anti-obesity properties of the novel peptide Ala-Gly-Leu-Gln-Phe-Pro-Val-Gly-Arg (AGL9), isolated from the enzymatic hydrolysate of *Allomyrina*
*dichotoma* larvae. To investigate the preventive effects of AGL9 against hepatic steatosis and its possible mechanisms of action, we established an nonalcoholic fatty liver disease (NAFLD) model by feeding C57BL/6 mice a high-fat diet. NAFLD mice were administered 100 mg/kg AGL9 and 60 mg/kg orlistat via gavage (10 mL/kg) for 5 weeks, followed by the collection of blood and liver tissues. We found that AGL9 normalized the levels of serum alanine aminotransferase, aspartate aminotransferase, triglyceride, total cholesterol, high-density lipoprotein, very low-density lipoprotein (LDL)/LDL, adiponectin, and leptin in these mice. Additionally, AGL9 activated the protein-level expression of 5′ AMP-activated protein kinase and acetyl-CoA carboxylase phosphorylation and the transcript-level expression of sterol regulatory element-binding protein-1c, fatty acid synthase, superoxide dismutase, glutathione peroxidase, glucocorticoid receptor, nuclear respiratory factor 2, tumor necrosis factor-α, interleukin-1β, interleukin-6, and monocyte chemoattractant protein-1 in hepatocytes. These results showed that AGL9 exhibited hepatoprotective effects by attenuating lipid deposition, oxidative stress, and inflammation via inhibition of AMPK/Nrf2 signaling, thereby reducing the production of hepatic proinflammatory mediators and indicating AGL9 as a potential therapeutic strategy for NAFLD.

## 1. Introduction

Nonalcoholic fatty liver disease (NAFLD) is a clinicopathological syndrome [1] characterized by pathological changes in hepatic fat metabolism, high triglyceride (TG) content in liver cells, and chronic inflammation [2]. Liver lobules are the focal points of the disease, which usually involves chronic benign processes; however, hepatic and systemic complications may occur in severe cases [1]. One explanation for NAFLD development is the “multiple-hit” hypothesis [3]. The contemporary pathophysiological model of NAFLD comprises parallel pathways exhibiting a dynamic cross-talk. The phenotypes range from simple steatosis to the histological manifestation of concurrent inflammation, a condition defined as non-alcoholic steatohepatitis (NASH) [4]. NAFLD is usually asymptomatic and can even be benign; however, ~20% of the patients with NAFLD develop fibrosis, cirrhosis, and hepatocellular carcinoma [5]. NAFLD and NASH have recently been recognized as hepatic manifestations of metabolic syndrome, with studies reporting a close relationship between NAFLD pathogenesis and inflammatory factors, oxidative stress, and lipid accumulation [6,7,8].

The nuclear respiratory factor 2 (Nrf2)-associated antioxidant system plays a key role in increasing the level of oxidative stress in NAFLD. Nrf2 activation effectively ameliorates NAFLD-induced inflammation, increased serum and intrahepatic lipid levels, and pathological changes in the liver in vitro and in vivo; this indicates that Nrf2 can serve as a promising target in NAFLD therapy [9]. Additionally, studies show that 5′ AMP-activated protein kinase (AMPK) plays an important role in NAFLD development. AMPK is extensively involved in regulating glucolipid metabolism in the body and has emerged as a hotspot for research on metabolic diseases and a current target for NAFLD drug research and development [10].

Although several drugs are being currently used for treating NAFLD/NASH, they tend to exhibit relatively narrow and subtle effects. Moreover, none of the drugs has been approved for NAFLD treatment, especially as the currently used drugs are often associated with serious side effects that can increase NAFLD-related risks [4]. Furthermore, the primary clinical recommendations for NAFLD prevention and treatment are usually lifestyle changes, healthy diets, and increased exercise [11]. Therefore, identifying effective approaches for NAFLD prevention and treatment has attracted extensive research interest.

Numerous studies have focused on the use of peptide drugs for NAFLD treatment [12,13,14]. Compared with other NAFLD-related drugs, peptide drugs are safe and well-tolerated and are characterized by a low molecular weight, thermal stability, low immunogenicity, low toxicity, and fewer side effects. Research has shown that most edible insects have medicinal value [15]. In some countries, insects are consumed as food supplements. Most edible insects can provide sufficient energy, have a high protein content and an ideal amino acid structure, a fatty acid composition, and are rich in some minerals and vitamins [16]. Though the current research on edible insects mainly focuses on their nutritional components, research on their pharmacological activity is limited.

We had previously purified and structurally characterized 23 peptides from *Allomyrina dichotoma* larvae using ultrafiltration and ion exchange chromatography and identified the active peptide Ala-Gly-Leu-Gln-Phe-Pro-Val-Gly-Arg (AGL9), which ameliorates abnormalities in glucolipid metabolism and exhibits pharmacological effects [17]. Given the therapeutic potential of *A. dichotoma* ALG9 in NAFLD, in the present study, we determined the function and mechanism of action of AGL9 with respect to reversing aberrant lipid metabolism and inflammation in the liver of NAFLD model mice fed a high-fat diet (HFD). To this end, we investigated AMPK/Nrf2 activity, as well as the expression of genes related to AMPK/Nrf2-regulated lipid synthesis and inflammation.

## 2. Materials and Methods

### 2.1. Materials

*A.**dichotoma* larvae, *Musca domestica* larvae, *Cetonia pilifera* larvae, *Tenebrio molitor* larvae, *Hermetia illucens* larvae, silkworm larvae, Orthoptera larvae, and *Velarifictorus aspersus* larvae were purchased in 2019 from Geoje Insect Farm (Gyeongnam-si, Korea). The larvae were fed fermented sawdust, and third-instar larvae were selected based on the size of their head capsules for subsequent experiments after being cleaned and frozen (−20 °C). Pepsin and trypsin were obtained from Sigma-Aldrich (St. Louis, MO, USA); neutral protease NP, pancreatin, AlphalaseTM NP, FoodPro alkaline protease, and promod 278P were obtained from Vision Biochem (Seoul, Korea); and promod 278P (PM); and alcalase and neutrase were obtained from Daejong Corporation (Seoul, Korea).

We used eight insect larvae (*A.*
*dichotoma* larvae, *Musca domestica* larvae, *Cetonia pilifera* larvae, *Tenebrio molitor* larvae, *Hermetia illucens* larvae, silkworm larvae, Orthoptera larvae, and *Velarifictorus aspersus* larvae) as raw materials for the extraction of biologically active peptides using a conventional hydrolytic transformation method. In accordance with a published method, we identified and collected AGL9 from 96 enzymatic hydrolysates of eight species of insects [17].

### 2.2. Cell Culture, Differentiation, and Staining

3T3-L1 preadipocytes (Korean Cell Line Bank, Seoul, Korea) were grown in 37 °C preadipocyte-expansion medium [Dulbecco’s modified Eagle medium (DMEM), 10% bovine calf serum, and 1% penicillin/streptomycin] in 12-well plates (for Oil Red O staining) until they reached post-confluency for 2 days. Differentiation was induced by modifying the medium to DMEM supplemented with 10% fetal bovine serum (FBS), 0.5 mM isobutyl methylxanthine, 0.5 mM dexamethasome, and 5 μg/mL insulin in the presence or absence of an enzymatic hydrolysis agent at the specified concentrations. After a 48-h induction, the separation medium was replaced with DMEM supplemented with 10% FBS and 5 g/mL insulin in the presence or absence of the indicated doses of Trypsin/EDTA solution. The culture medium was then replaced every other day with DMEM containing 10% FBS in the presence or absence of the required doses of enzymatic hydrolysis solution up to day 8, at which time the preadipocytes had matured and were spherical and densely packed with fat droplets [17].

Cellular lipid content was measured using a previously described method [18]. Cells were fixed with 10% formalin at room temperature (25 °C) for 60 min and washed with 60% isopropanol, after which the cells were stained with Oil Red O (0.4% Oil Red O dye in 60% isopropanol) for 20 min. Images were obtained using a binocular microscope (DMi1; Leica Microsystems, Wetzlar, Germany). The dye stored in the cells was eluted with isopropanol and quantified using a microplate reader to calculate the optical absorbance at 520 nm.

### 2.3. Dosage Information/Regimen

We obtained male C57BL/6 mice (4-weeks old) from Nara Bio Animal Center (NARA Biotech, Seoul, Korea). The mice had free access to water throughout the study. After 1 week of acclimation, mice were randomly divided into two groups, i.e., the control group (*n* = 8), which was fed a normal diet (CON; 10% kcal fat content; D12450K, Research Diets, Inc., New Brunswick, NJ, USA); and the experimental group (*n* = 30), which was fed HFD (60% kcal fat content; D12492; Research Diets, Inc.) [18]. Food consumption was monitored daily, and body weights were determined twice a week. After 5 weeks, mice with weights 20% more than those of the control group were randomly split into three groups (*n* = 8), i.e., HFD-fed mice administered saline (NAFLD); HFD-fed mice administered 100 mg/kg AGL9 (NAFLD+AGL9); and HFD-fed mice administered 60 mg/kg orlistat (NAFLD+orlistat; 10 mL/kg) for 5 weeks. Food consumption was reported three times each week, and the body weight of each mouse was assessed once every week. Orlistat-treated mice were used as a positive control.

After 10 weeks, mice were fasted overnight and euthanized. Blood was collected from the hearts, and serum was acquired by centrifugation (1000× *g*, 4 °C, 20 min). Liver tissues were then rapidly extracted and weighed. Following image acquisition, the tissues were stored at −80 °C until further use. Liver tissue sections were fixed for histologic examination in 10% formaldehyde. This study was authorized by Konkuk University’s Institutional Animal Care and Use Committee, and every attempt was made to reduce the pain and number of animals involved in the study (KU20018).

### 2.4. Histological Evaluation

For histological examination, liver tissues were fixed in 10% formalin and embedded in paraffin, followed by sectioning to a thickness of 4 µm and staining with hematoxylin and eosin (H&E). Frozen liver tissues were fixed in 10% formalin for 10 min and then washed with fresh water. After a few rounds of cleaning with distilled water, the sections were immersed in isopropanol for 20 s to 30 s. After staining the sections with Oil Red O for 15 min to 20 min, they were washed with distilled water. After 40 s of staining with hematoxylin, the tissue sections were immersed in tap water for 5 min. The sections were then imaged at 200× magnification using a DMi1 microscope (Leica Microsystems). Adipocyte sizes were assessed using ImageJ 1.8.0 (NIH, Bethesda, MD, USA).

### 2.5. Immunohistochemistry

Immunohistochemistry was performed as previously described [17]. Briefly, mouse liver tissue sections were permeabilized for 30 min at room temperature (25 °C) in Tris-buffered saline (TBS) containing 0.3% Triton X-100, boiled for 20 min in Tris-EDTA (pH 9.0), and blocked for 1 h with 5% goat serum prepared in TBS (pH 7.4) to minimize nonspecific binding. For immunohistochemistry, the tissue sections were probed overnight with the anti-sterol regulatory element-binding protein-1c (SREBP-1c) and anti-fatty acid synthase (FAS) antibodies.

### 2.6. Biochemical Analysis

An automated analyzer (Abaxis VETSVAN VS2 Chemistry Analyzer; Abaxis, Union City, CA, USA) was used to determine the levels of alanine aminotransferase (ALT) and aspartate aminotransferase (AST) in the serum. A fast blood lipid analyzer was used to determine very low-density lipoprotein (VLDL)/LDL, total cholesterol (TC), high-density lipoprotein (HDL), and TG levels (Lipid Pro; OSANG Healthcare, Anyang, Korea). Adiponectin levels were determined using an enzyme-linked immunosorbent assay (ELISA) kit specific for mouse adiponectin (Invitrogen, Thermo Fisher Scientific, Carlsbad, CA, USA). Leptin was measured using an ELISA kit specific for leptin (MERCK, Darmstadt, Germany).

### 2.7. Evaluation of the Transcript-Level Expression of Target Genes Using Quantitative Real-Time PCR (qPCR)

As previously described [18], total RNA was extracted from the liver for qPCR. The primers used for qPCR were synthesized by Enotech Co. (Daejeon, Korea) (Table 1). Target gene expression was normalized to that of *glyceraldehyde 3-phosphate dehydrogenase*, and the 2^−ΔΔCt^ method was used to determine relative expression [18].

### 2.8. Western Blotting

Total protein content of lysed liver tissue was extracted in cold radioimmunoprecipitation assay lysis buffer, followed by three rounds of centrifugation at 18,894× *g* for 20 min at 4 °C. A BCA protein assay kit (Thermo Scientific, Rockford, IL, USA) was used to quantify the protein levels, and western blotting was performed as previously described [18].

### 2.9. Statistical Analysis

One-way analysis of variance was used to determine differences between groups, and data were represented as the mean ± standard error of the mean. Turkey’s post hoc test was performed using GraphPad Prism software (v.8.0; GraphPad Software, La Jolla, CA, USA); *p* < 0.05 was considered significant.

## 3. Results

### 3.1. The A. dichotoma Larva Enzymatic Hydrolysate Improves Lipid Deposition in 3T3-L1 Cells

An *A. dichotoma* larva PM enzymatic hydrolysate was selected from 96 enzymatic hydrolysates of larvae from eight insect species, as they resulted in the highest resistance to lipid deposition (Figure 1). Numerous natural active peptides were extracted from the *A. dichotoma* larvae enzyme hydrolysate. AGL9 is a short peptide comprising nine amino acids (molecular weight, 943 Da). Compared with other natural active peptides, AGL9 has advantages, such as a small molecular weight, strong anti-lipid‒deposition activity, and high stability [17].

### 3.2. Effect of AGL9 on Liver Injury in NAFLD Mice

According to human obesity and weight standards, a body mass > 10% than the ideal level is regarded as overweight and >20% is regarded as obese. Therefore, when the mean body mass of mice in the HFD group exceeded 1.2-fold, the mean body mass of the mice in CON group, we considered the NAFLD model to be successfully established. After 5 weeks of feeding, the average body weight of mice fed a normal diet was 24.86 g, whereas that of mice fed HFD was 32.28 g, which was 1.29-fold higher than that of mice in the CON group. NAFLD mice were then administered saline, orlistat, or AGL9 for 5 weeks, which resulted in an increase in body weight in all groups, suggesting normal growth (Figure 2). Furthermore, compared with weight increases in mice in the NAFLD group (20.21 g) after 10 weeks of feeding, mice in the NAFLD+AGL9 group showed an increase of only 11.30 g in body weight (*p* < 0.05), suggesting that AGL9 inhibited the body-weight increases in NAFLD mice.

We then used Oil Red O and H&E staining to identify pathological alterations in the liver tissue. Compared with those from the CON group, liver tissues from the NAFLD group demonstrated a higher degree of diffuse steatosis, which was significantly reduced in NAFLD+AGL9 mice along with lower levels of inflammatory cell infiltration. Additionally, ALT and AST levels—commonly used as markers of liver function—were substantially higher in the NAFLD group relative to those in the CON group, indicating the presence of liver injury in NAFLD mice (Figure 3). However, NAFLD+AGL9 mice showed substantially lower ALT and AST levels relative to NAFLD mice. These findings indicated that AGL9 was beneficial with respect to reversing HFD-induced liver damage.

### 3.3. Effect of AGL9 on Lipid Metabolism in NAFLD Mice

Evaluation of blood metabolic indices in each group revealed substantial increases in TC, TG, and VLDL/LDL levels relative to those in the CON group, whereas these increases were not observed in NAFLD+AG mice (*p* < 0.05). Similarly, serum HDL levels substantially decreased in NAFLD mice (relative to those in the CON group) were elevated in NaFLD+AGL9 mice to levels similar to those in the CON group (Table 2).

### 3.4. Effect of AGL9 on AMPK Signaling in NAFLD Mice

We then determined changes in the phosphorylation levels of acetyl-CoA carboxylase (ACC) and AMPK. The results revealed significant reductions in these phosphorylation levels in NAFLD mice relative to those in the CON group (*p* < 0.05), whereas NAFLD+AGL9 mice exhibited AMPK and ACC phosphorylation levels similar to those in the CON group (Figure 3A). Additionally, we determined SREBP-1 and FAS expression in the liver tissues from each group to evaluate the effect of AGL9 on lipid synthesis. Compared with those in NAFLD mice, FAS and SREBP-1 levels in NAFLD+AGL9 mice were significantly decreased (*p* < 0.05) (Figure 4).

### 3.5. Effects of AGL9 on Oxidative Stress and Inflammation in the Liver of NAFLD Mice

We then measured the levels of superoxide dismutase (SOD), glutathione peroxidase (GPx), and glucocorticoid receptor (GR) in the liver (common indexes of oxidative stress). The results showed that mRNA levels of *Sod*, *Gpx*, and *Nr3c1* (encoding GR) in NAFLD were substantially lower than those in the CON group, whereas those in NAFLD+AGL9 mice were similar to those in the CON group. Additionally, we found similar variations in *Nrf2* mRNA levels, including their recovery following AGL9 treatment (Figure 5). These findings suggested the ability of AGL9 to reduce oxidative stress in the livers of NAFLD mice. Furthermore, we observed elevated expression of the inflammatory markers interleukin 1b (*Il-1b*), tumor necrosis factor a (*Tnfa*), *monocyte chemoattractant protein-1* (*Mcp1*), and *Il6* in NAFLD mice (relative to that in the CON group); these levels recovered following AGL9 treatment, suggesting that AGL9 effectively reduced liver inflammation in NAFLD mice.

## 4. Discussion

NAFLD is a hepatic metabolic syndrome characterized by lipid overaccumulation, inflammation, and hepatic parenchymal cell lipidation [3]. In this study, we established an NAFLD mouse model recapitulating the pathophysiological changes observed in human NAFLD [19]. Although many NAFLD-related drugs are clinically efficacious, they are all associated with serious side effects and occasionally induce the development of drug resistance [4,20]. Therefore, the exploration of therapeutic agents exhibiting low toxicity and high efficiency, including those from natural foods, has become a popular area of research. AGL9, an active peptide from *A. dichotoma* larvae, improves insulin resistance and dyslipidemia in mice with HFD-induced obesity [19], thereby reducing abnormal hepatic lipid accumulation. In the present study, we found that AGL9 administration improved the NAFLD phenotype in mice.

H&E and Oil Red O staining revealed that NAFLD mice presented significant hepatic steatosis, and that AGL9 intake effectively inhibited fat deposition. These data indicated that AGL9 might prevent liver steatosis by reducing the lipid content. Lipid accumulation in liver parenchymal cells and subsequent toxicity can result from excessive inflammation and liver damage [21]. Moreover, HFD can cause an increase in free fatty acids (FFAs) in serum, with the continuous accumulation of FFAs causing liver damage, increasing cell-membrane permeability and the release of ALT and AST into the circulation, where they are frequently used as important markers of the degree of liver damage [22]. In the present study, we observed increases in serum ALT and AST levels accompanied by increases in TC, TG, and LDL levels and a decrease in HDL content in NAFLD mice. These findings confirmed the establishment of the NAFLD model, with subsequent findings revealing that ALG9 treatment decreased ALT and AST levels and alleviated HFD-induced liver damage.

NAFLD involves the deposition of TGs in liver cells, which is caused by an imbalanced intake and expenditure of TGs and FFAs due to dysfunctional lipid metabolism [3]. The primary characteristics of HFD-induced NAFLD include increased TC, TG, and LDL levels and reduced HDL levels, which also lead to inefficient lipid metabolism [23]. A previous study showed that reducing TC, TG, and LDL levels can effectively alleviate NAFLD [24]. Adiponectin is a cytokine expressed and secreted by adipose tissue and that exerts anti-inflammatory effects, increases fatty acid β-oxidation, and reduces lipid accumulation in liver cells [25]. In the present study, we found that AGL9 administration substantially decreased adiponectin levels and rescued the NAFLD phenotype, suggesting its potential as a treatment strategy for lipid-metabolism disorders.

NAFLD development and lipid metabolism are closely associated with AMPK, SREBP-1c, and ACC signaling pathways. The AMPK/ACC signaling pathway is important for regulating energy metabolism, lipid synthesis, and oxidation processes [26], with AMPK specifically involved in maintaining energy homeostasis and regulating lipid metabolism [10]. Activation of AMPK by phosphorylation indirectly regulates the expression of adipogenic genes, such as FAS and ACC, by downregulating SREBP-1c expression, thereby inhibiting TG synthesis [27]. Interestingly, AMPK can promote ACC phosphorylation to reduce ACC activity, thereby inhibiting adipogenesis and activating fatty acid oxidation in mitochondria [28]. Therefore, AMPK activation might represent a therapeutic target for alleviating NAFLD. SREBPs are membrane-bound transcription factors involved in lipid metabolism, and SREBP-1c positively regulates the expression of fatty acid synthases, including FAS and ACC, thereby regulating TG and FFA synthesis and intracellular cholesterol and lipid metabolism [29]. FAS is an important enzyme involved in fatty acid synthesis in liver cells, with increased FAS activity involved in esterification of fatty acids and increased lipid deposition in animals [27]. Other studies report that inducing AMPK phosphorylation and downregulating SREBP-1c expression and its target genes significantly reduces lipid deposition [30]. In the present study, we identified significantly attenuated AMPK phosphorylation in NAFLD mice. Notably, AGL9 treatment significantly increased both ACC and AMPK phosphorylation, thus downregulating the expression of their downstream targets SREBP-1c and FAS and reducing lipid synthesis in liver cells. These results indicated that AGL9 might activate AMPK signaling to inhibit lipid synthesis and thereby improve liver-specific lipid metabolism in NAFLD.

During NAFLD development, oxidative stress and lipid peroxidation represent secondary promoters of disease progression. During NAFLD progression, fatty liver disease develops into steatohepatitis, which eventually causes inflammation, necrosis, and liver fibrosis. Inflammation and oxidative stress are interdependent and occur during NAFLD onset [3]. During the early stage of NAFLD, phagocytes activated during inflammation generate excess reactive oxygen species to kill pathogens under the action of proinflammatory factors. At the same time, phagocytes induce local oxidative stress and tissue damage, further activating inflammation-related signaling pathways and accelerating the development of inflammation [31]. In the present study, we performed qPCR analysis of oxidative-stress- and inflammation-related gene expression in liver tissues The most common oxidative stress markers include SOD, GPx, and GR [32], which represent enzymes involve in antioxidant defense against harm by suppressing free radical production, scavenging free radicals, repairing oxidative damage, and activating other antioxidant enzymes. Additionally, these enzymes can effectively inhibit the formation of lipid peroxidation reactants [33]. The qPCR results showed decreased expression of all three enzymes in liver tissue from NAFLD mice; however, AGL9 restored these levels, suggesting that AGL9 might alleviate HFD-induced oxidative stress by improving the antioxidant capacity of liver cells.

Nrf2 controls the transcription of antioxidant factors and is involved in cellular resistance to oxidative stress [34]. Numerous studies demonstrate that AMPK can activate Nrf2 [35,36,37]. In response to oxidative stress, Nrf2 is produced and translocates to the nucleus, where it regulates the production of proinflammatory target genes, such as *IL-1β, TNF-α, MCP-1*, and *IL-6* [38]. In the present study, AGL9 treatment increased the levels of *Nrf2* mRNA and reduced the levels of *Il1b, Tnfa, Mcp1*, and *Il6* in liver tissue, suggesting that AGL9 can activate the Nrf2 signaling pathway. Taken together, these results indicate that AGL9 has the potential to reduce oxidative stress and inflammation in patients with NAFLD.

Further structural characterization of AGL9 is necessary to elucidate the structural and functional relationships between AGL9 and metabolic pathways. The mechanisms associated with lipid-lowering effects are diverse, and the present study elucidated the hepatoprotective mechanism of AGL9 through the AMPK/Nrf2 pathway; however, other pathways should be investigated to determine its possible roles in improving liver injury. Moreover, our future research will establish an experimental basis and structural evidence supporting the development of novel functional nutraceuticals for obesity prevention and potential food additives for NAFLD treatment.

## 5. Conclusions

These findings suggest that ALG9 activates AMPK/Nrf2 signaling to regulate lipid levels and suppress lipid metabolism and oxidative stress in an NAFLD mouse model. Additionally, ALG9-mediated inhibition of the expression of fat synthesis-related genes, such as FAS and SREBP-1c, reduced lipid deposition, improved the degree of steatosis, and exhibited lipid-lowering effects. Furthermore, ALG9 treatment attenuated the expression of inflammatory factors that protected the liver from oxidative stress. These findings offer insight into the potential application of AGL9 for the treatment of NAFLD.

## Figures and Tables

**Figure 1 foods-10-01973-f001:**
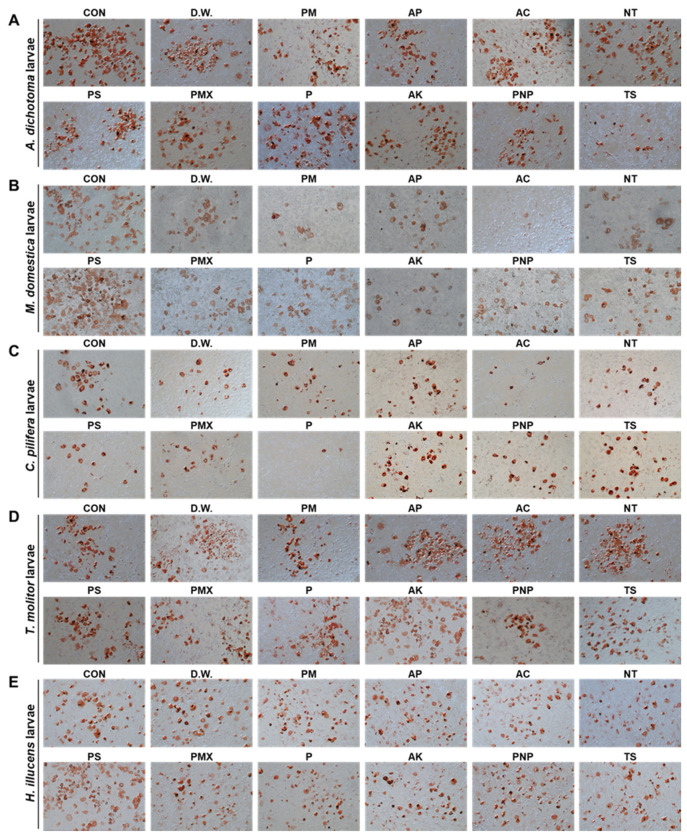
The anti-obesity effect of insect enzyme hydrolysate. Oil Red O staining of 3T3-L1 cells treated with the enzyme hydrolysate of (**A**) *Allomyrina*
*dichotoma* larvae, (**B**) *Musca domestica* larvae, (**C**) *Cetonia pilifera* larvae, (**D**) *Tenebrio molitor* larvae, (**E**) *Hermetia illucens* larvae, (**F**) silkworm larvae, (**G**) Orthoptera larvae, and (**H**) *Velarifictorus aspersus* larvae. As a control, untreated 3T3-L1 cells were used. (**I**) Resistance to lipid accumulation. CON, control; D.W., double-distilled water; PM: promod 278P; AP: alphalase NP; AC: alcalase; NT: neutrase; PMX: PS pepsin protamex; P: pancreatin; AK: Food Pro alkaline protease; PNP: protease NP; TS: trypsin.

**Figure 2 foods-10-01973-f002:**
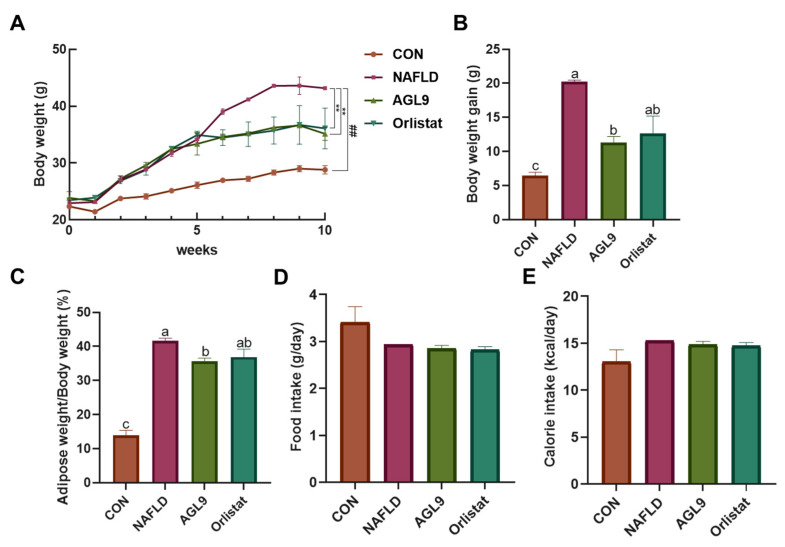
Effects of AGL9 on obesity. (**A**) Body weight, (**B**) body-weight gain, and (**C**) adipose weight of NAFLD mice. AGL9 reduces lipid accumulation in the adipose tissue. Diet-related effects on (**D**) food intake and (**E**) calorie intake. Data represent the mean ± standard error of the mean. Different lowercase letters represent a significant difference (*p* < 0.05) according to Dunnett’s multiple range test.

**Figure 3 foods-10-01973-f003:**
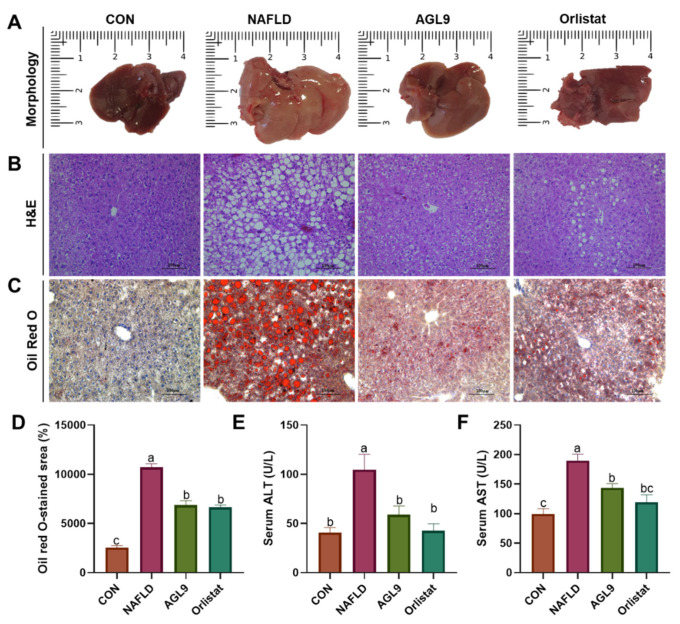
Effects of AGL9 on liver injury. (**A**) Liver morphology in NAFLD mice. AGL9 reduces lipid accumulation in the liver. Histological examination of liver and epididymal adipose tissues using (**B**) H&E and (**C**) Oil Red O staining. (**D**) Mean liver adipocyte area (μm^2^). Determination of serum levels of (**E**) ALT and (**F**) AST. Data represent the mean ± standard error of the mean. Different lowercase letters represent a significant difference (*p* < 0.05) according to Dunnett’s multiple range test. H&E, hematoxylin and eosin; HFD, high-fat diet; ALT, alanine aminotransferase; AST, aspartate aminotransferase.

**Figure 4 foods-10-01973-f004:**
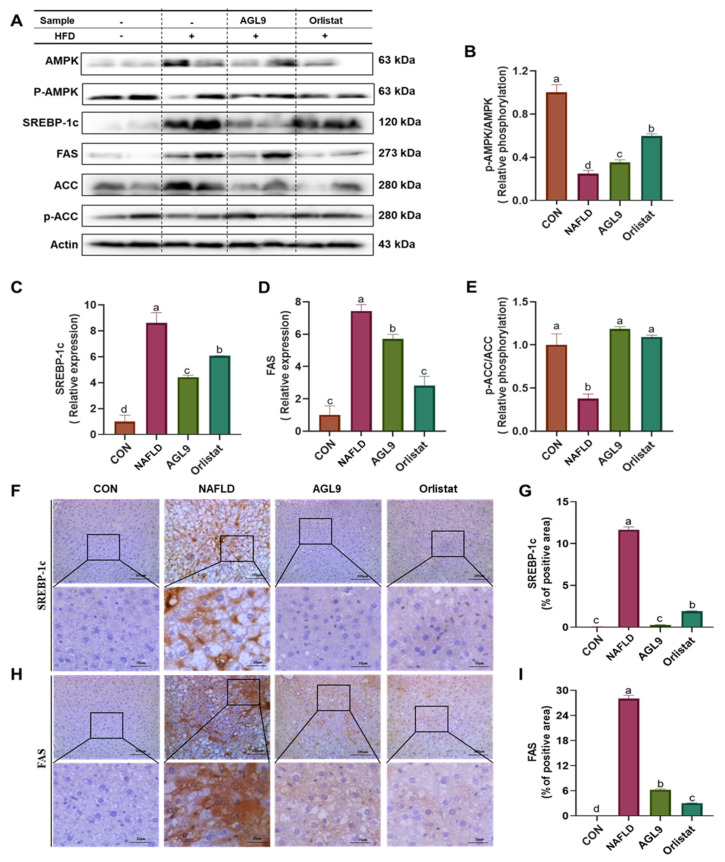
Effect of AGL9 on AMPK signaling in NAFLD mice. (**A**) Immunoblot analysis of (**B**) phospho-AMPK/AMPK, (**C**) SREBP-1c, (**D**) FAS, and (**E**) phospho-ACC/ACC in the liver tissue. Protein levels were normalized to those of β-actin. Immunostaining of liver sections for (**F**) SREBP-1c and (**H**) FAS (original magnification, 200×; scale bars = 100 µm). Arrows indicate crown-like structures. Areas of positive staining for (**G**) SREBP-1c and (**I**) FAS in liver tissues. Data represent the mean ± standard error of the mean. Superscript lowercase letters represent a *p* < 0.05 according to Dunnett’s multiple range test.

**Figure 5 foods-10-01973-f005:**
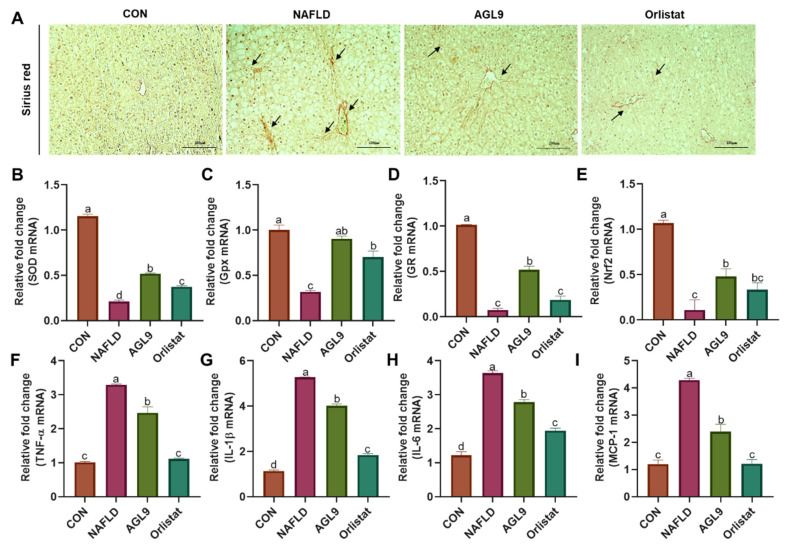
Staining of liver tissue for markers of oxidative stress and inflammation. (**A**) Representative Sirius red-stained liver sections. Black arrows identify areas of fibrosis. Real-time qPCR quantification of mRNA levels in mouse liver tissue. Graphs show mRNA expression of (**B**) *Sod*, (**C**) *Gpx*, (**D**) *Nr3c1* (encoding GR), (**E**) *Nrf2*, (**F**) *Tnfa*, (**G**) *Il1b*, (**H**) *Il6*, and (**I**) *Mcp1*. Data represent the mean ± standard error of the mean. Superscript lowercase letters represent a *p* < 0.05 according to Dunnett’s multiple range test.

**Table 1 foods-10-01973-t001:** Primers used for qPCR.

Gene Name		Sequence
*Atgl*	Forward	5′-GAC CTG ATG ACC ACC CTT TCC-′3
	Reverse	5′-TGC TAC CCG TCT GCT CTT TCA-′3
*Sod*	Forward	5′-CAA TGG TGG GGG ACA TAT TA-′3
	Reverse	5′-TTG ATA GCC TCC AGC AAC TC-′3
*Gpx*	Forward	5′-ACA TTC CCA GTC ATT CTA CC-′3
	Reverse	5′-TTC AAG CAG GCA GAT ACG-′3
*Nr3c1*	Forward	5′-CGG CGA TCT CCA CAG CAA TG-′3
	Reverse	5′-ACC GCT CCA CAC ATC CTG ATT G-′3
*Nrf2*	Forward	5′-AGC ACA TCC AGA CAG ACA CCA GT-′3
	Reverse	5′-TTC AGC GTG GCT GGG GAT AT-′3
*Tnfa*	Forward	5′-AAG CCT GTA GCC CAC GTC GT-′3
	Reverse	5′-GGC ACC ACT AGT TGG TTG TC-′3
*Il1b*	Forward	5′-AAC CAA GCA ACG AVA AAA TA-′3
	Reverse	5′-AGG TGC TGA TGT ACC AGT TG-′3
*Il6*	Forward	5′-CCG GAG AGG AGA CTT CAC AG-′3
	Reverse	5′-GGA AAT TGG GGT AGG AAG GA-′3
*Mcp1*	Forward	5′-TGA TCC CAA TGA GTA GGC TGG AG-′3
	Reverse	5′-ATG TCT GGA CCC ATT CCT TCT TG-′3
*Gapdh*	Forward	5′-GCA CAG TCA AGG CCG AGA AT-′3
	Reverse	5′-GCC TTC TCC ATG GTG GTG AA-′3

**Table 2 foods-10-01973-t002:** Effects of AGL9 on serum markers.

	Parameter	CON	NAFLD	AGL9	Orlistat
Serum	TG (mg/dL)	108.33 ± 2.52 ^c^	232.60 ± 3.42 ^a^	138.76 ± 6.25 ^b^	117.92 ± 4.72 ^c^
	TC (mg/dL)	96.06 ± 4.72 ^b^	119.65 ± 6.03 ^a^	120.46 ± 4.32 ^a^	119.26 ± 2.80 ^a^
	HDL (mg/dL)	79.81 ± 4.79 ^a^	47.35 ± 6.03 ^b^	77.81 ± 1.23 ^a^	75.05 ± 2.80 ^a^
	LDL/VLDL (mg/dL)	16.25 ± 2.64 ^c^	72.30 ± 2.20 ^a^	42.65 ± 5.57 ^b^	44.21 ± 4.39 ^b^
	Adiponectin (μg/mL)	53.45 ± 4.46 ^b^	84.19 ± 5.52 ^a^	66.71 ± 4.26 ^b^	67.43 ± 5.11 ^b^
	Leptin (ng/mL)	4.80 ± 1.68 ^c^	45.40 ± 3.14 ^a^	22.08 ± 2.74 ^b^	20.08 ± 2.67 ^b^

Abbreviations: HFD, high-fat diet; TGs, triglycerides; TC, total cholesterol; HDL, high-density lipoprotein cholesterol; VLDL, very low-density lipoprotein cholesterol; LDL, low-density lipoprotein cholesterol; CON, control; AGL9, Ala-Gly-Leu-Gln-Phe-Pro-Val-Gly-Arg; NAFLD, nonalcoholic fatty liver disease. Data represent the mean ± standard error of the mean. Superscript lowercase letters represent a *p* < 0.05 according to Dunnett’s multiple range test.

## Data Availability

The data supporting the findings of this study are available from the corresponding author upon request.

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
