# Peer review of "AGL9: A Novel Hepatoprotective Peptide from the Larvae of Edible Insects Alleviates Obesity-Induced Hepatic Inflammation by Regulating AMPK/Nrf2 Signaling"

_foods, 2021, doi:10.3390/foods10091973_

Round 1
Reviewer 1 Report
Fan et al. present findings of a novel peptide, named AGL9, which is demonstrated to lead to interesting improvements diet-induced hepatic steatosis and inflammation. While I am intrigued by the results of this paper, I feel that the description of the findings is not in line with their actual findings.
Major comments:
- The authors refer to an NAFLD mouse model, by feeding C57BL/9 mice a HFD diet. I, however, raise questions whether this model resembles all feature as observed in human NAFLD (steatosis, inflammation and fibrosis).
- The two-hit hypothesis, which the authors claim in line 45, has been outdated since 10-15 years. I strongly advice to change this to the current view in the field (multiple hit-hypothesis).
- In relation to the rest of the paper, the authors should provide more information on what NAFLD actually concerns (spectrum, stages, etc). They refer to ‘beginning stage of NAFLD’, but this cannot be clear for a general reader what is meant with this.
- In the introduction, the referenced studies on exendin-4 and liraglutide come out-of-the-blue, similar to the mention of the statement of the FAO (line 69). I feel the entire introduction needs restructuring. It does not feel to me as an introduction to the rest of the paper, but rather point-by-point statements, leading to the research question. (I would for example really like to read more about the source of ALG9)
- In the introduction, the authors claim as goal: to determine the function and mechanism of ALG9…’; however, no mechanistic studies are undertaken to show how ALG9 functions…
- I do not understand figure 1; what do the abbreviations mean? (CON; DW; PM; etc) What was exactly done? I feel I cannot assess the validity of this figure.
- Figure 2 is very impressive… however, I do not see improvements in inflammatory cells based on the HE staining (only changes in lipid metabolism are clear to me from these data). The claim on inflammation should therefore be deleted (or additional data need to be provided in this figure).
- While the inflammatory data in figure 5 are convincing, to fully claim improvements in liver inflammation, the authors should show this inflammation also at histological level (as this is the standard when assessing inflammation in patients).
- Unfortunately, the entire discussion of the authors is centered around the ‘two-hit hypothesis’. As I mentioned before, this view has been outdated for at least a decade. I can therefore only support publication if this is rewritten.
- Linked to my previous comment: Figure 5 should therefore be deleted. Also because the authors do show any causal link between AGL9 and Nrf2; only associations are observed in this paper
- Something I would really like to see in the discussion is how the authors envision to continue with this peptide research + how such peptide can be implemented as treatment for patients.
- English editing is strongly recommended. I even feel that I cannot agree on publication without strong improvements in scientific English
Minor comments:
Line 35: ‘HFD-induced NAFLD’ à NAFLD is always HFD-induced; the first part seems therefore redundant
Line 42: ‘Liver lobules are central to the disease, which usually involves chronic benign processes’; it is unclear to me what the authors mean with this sentence. I advice to rephrase
Line 55-57: ‘Therefore, although there are currently no authorised medicines for the treatment of NAFLD, increasing numbers of drugs are being tested to treat this disease’
Line 83: the authors refer to NAFLD patients in their objective, but no human data is presented in the paper…
Line 126: The content of the HFD should be provided in detail (content-wise).Was cholesterol added to the diet or not? Did the authors make the diet themselves or did they buy it?
Line 225: ‘but’ needs to be deleted
Figure 4B-4I: in several graphs, the control group is not averaged at 1. For proper interpretation of these graphs, this should be done
Reviewer 2 Report
The authors investigated the anti-oxidative and anti-inflammation effects of peptide from edible larvae in the model of NAFLD in mice. One of the major concerns is that the obtain of peptide is confused for readers, and the figure legends is too simple so that it seems confused for readers in terms of the design and the purpose of the experiments. The major and minor concern are listed as follow:
- In materials section, how did authors obtain active peptide from larvae? Describe the way to obtain the AGL9 from eight larvae.
- What is the criterium to choose 100mg/kg AGL in mice?
- Line 192-194, no evidence to support this sentence.
- Figure 1, describe the figure 1 clearly, it is confused for readers in terms of the names of D.W., AP, AC….
- In section 3.2 and 3.3, how about the body weight change among different groups, liver weight, and white/brown adipose tissue weight?
- Figure 4, Sirius red indicates the fibrosis of liver, how about related gene expressions, such as collagen, Mmp and Timp families…. What does black arrow mean in figure 4, describe it in figure legend.
- Figure 4, how about the expression levels of inflammation related markers in blood, like Mcp-1, IL-6…
- List the primers in the manuscript.
Minors:
- Line 67, I donot think “there is no experimental evidence for the medicinal value of edible insects”, For instance, research about edible insects in terms of their antioxidation, anti-inflammation activities….10.5455/jcmr.20190130100319
- Line 55, drugs used to treat NAFLD include ursodiol, actos, and pioglitazone, and so on.
- Line 136, change 3000 rpm to g.
- Line 192, molecular weight 943 Da?
Round 2
Reviewer 1 Report
While all my previous comments were considered, I still feel that the level of scientific English is rather low. (I am therefore surprised that the text was edited by a native English speaker…) In order to support publication, English editing is required.
Reviewer 2 Report
My concerns about the manuscript are resolved, try to improve the language before publication.
